# Using Community of Inquiry (CoI) to Facilitate the Design of a Holistic E-Learning Experience for Students with Visual Impairments

**Sindile A. Ngubane-Mokiwa** [1,*] **and Simon Bheki Khoza** [2]

1   Open Distance Learning Research Unit, College of Education, University of South Africa, Pretoria 0001, South Africa

2   Curriculum Studies, School of Education, College of Humanities, University of KwaZulu-Natal, Durban 4000, South Africa; khozas@ukzn.ac.za

*   Correspondence: mokiwsa@unisa.ac.za

**Abstract:** Information and Communication Technology (ICT) tools provide an ideal mechanism by which students can interact closely with their lecturers in an open distance learning (ODL) context. This is especially true for students with disabilities who require access through alternative formats. This paper demonstrates teaching and learning practices in an ODL institution in South Africa, with a focus on the indispensable role of lecturers and tutors in an online learning setting. The paper employs the Community of Inquiry, which sees the effective online learning environment through three elements: cognitive, social, and teaching presence. The findings shed light on the use of vision-based approaches in course design; limited implementation of open-access policies, and the academic faculty's lack of knowledge on how to facilitate inclusive learning. The paper concludes by presenting a proposed student-centred framework that seeks to facilitate inclusive teaching and learning towards positive and inclusive learning experiences for students. The proposed framework could be beneficial during pandemic situations.

**Keywords:** community of inquiry; educational experience; cognitive; presence; social teaching

## 1. Introduction

Founded in 1873 as the University of Cape of Good Hope, changed to the University of South Africa (UNISA) in 1916, since 1946, this institution has offered distance education over these 145 years using various forms of technology. Before the advent of emerging technologies, open distance learning (ODL) was facilitated by paper-based tutorial letters, audio-tapes, and audio compact discs. However, as the number of students increased, it became increasingly difficult to interact with students through traditional technologies [1]. The open distance and e-learning (ODeL) model introduced in 2009 focused on delivering quality higher education to students [2] through the use of emerging technologies, was an extension of ODL. Owing to the distance between the academics and the students, ODeL strives to use efficient technologies to support students. UNISA, thus, has introduced innovative models of teaching and learning as part of its student-support initiatives, which entail the use of online learning platforms. However, the developed models lack the identification of those whose needs are addressed by these models [3]. As a result, the needs of students with disabilities are not considered in most of the ODeL models, because they are unknown [4,5]. Therefore, this study explores the use of the community of inquiry (CoI), shaped by Garrison, Anderson and Archer [6], to facilitate the design of a holistic e-learning experience for students with visual impairments. It was guided by the following research questions: (1) What are the learning needs of students with visual impairments? (2) How can students with visual impairments be taught in an ODeL environment? (3) How can the e-learning environment be designed to support students with visual impairments

in times of pandemics which impose restrictions? (4) Why must students with visual impairments be supported and taught in particular ways? It is hoped that this paper will contribute towards better use of CoI to mediate positive learning experiences for students with disabilities.

## 2. Literature Review

On the 23 March 2020, UNISA, like all other universities in South Africa, was instructed by the state president, Cyril Ramaphosa, to shut down (national lockdown) all their activities, in order to contain the spread of the coronavirus of 2019 (COVID-19) [7]. Thus, the COVID-19 national lockdown forced UNISA to innovate new approaches to accommodate the needs of students with disabilities, facilitating teaching and learning during national lockdown. The adapted model has the potential to reduce the teaching and learning distance between the university and the students, the academics and the students, the learning content and the students, and amongst the students themselves (from both advantaged and disadvantaged communities).

The ODeL model, which includes multipurpose information and communication technology (ICT) Labs, is best suited to accommodate the diverse learning needs of students with disabilities, owing to its design and plan. In this way, the multipurpose ICT Labs accommodate physical, neurological, and intellectual disabilities [6]. To this effect, Hollier [8] (p. 10) asserts that ICT brings "hope" to students with visual impairment, particularly the completely blind. Through ICT, students with visual impairment can access various kinds of information, learn independently, and register online, much as they can bank and shop online. If the learning environment is not designed to fulfil the students' learning needs, then it does not serve its purpose [5,9]. This is normally because students with disabilities do not always have access to a learning environment that supports or strives to facilitate the attainment of their learning goals based on their needs. In this regard, Eligi and Mwantimwa [10] postulate that lack of adequate learning support for students with visual impairment is the reason for there being few students with disabilities in higher education institutions. Even fewer of such students complete their degrees. This assertion supports Fraser and Maguvhe [11] (p. 1), that students with visual impairments are "the most vulnerable individuals in terms of the learning mediation". In addition, Amory [12]; Czerniewicz, and Brown [13] point out that ODeL instructors lack the skills to design for interactive learning through relevant approaches, which further affects visually impaired students. Livingstone [14] (p. 11), accordingly, warns that "the mere presence of ICT does not guarantee effective learning but rather enhances the students' learning experience", a point also raised by researchers such as Means, Toyama, Murphy, Bakia and Jones [15] (p. 2). Consequently, it has been suggested that "open and distance education educators should track and trace the learning habits and behaviours of their distance students in order to design and deliver their courses efficiently" [16] (p, 86).

In designing online learning, ODeL educators should consider spatial and temporal factors that might affect interactions, relevant content and interactive media, including media which facilitates collaboration [17,18]. It has been found that the use of the Internet, as a learning platform, makes learning and studying resources accessible, not merely in libraries, but also at home and at work [19,20]. The questions, then, are: what are the learning needs of students with visual impairments? How can students with visual impairments be taught in an ODeL environment? How can the e-learning environment be designed to support students with visual impairments in times of pandemics which impose restrictions? Why must students with visual impairments be supported and taught in particular ways?

The aim of this paper, therefore, is to use the elements of the CoI to facilitate the design of a holistic e-learning experience for students with visual impairments, one which will cater for various interactions, while making resources accessible. To contribute to this debate, this paper discusses other models that have been used to inform e-learning design. These designs use the CoI, a model that has been confirmed through content investigation

and other qualitative and quantitative studies, by contextualising it to UNISA. The CoI has been referenced by many researchers [7,21–26], and is ostensibly the most famous model regarding inclusive online learning.

*The Community of Inquiry as an Inclusive Model for ODL*

To give a psychological schema to pondering teaching and learning, Garrison, Anderson, and Archer [1] advanced a theoretical model of online-based learning (e-learning) called the Community of Inquiry (CoI) (Figure 1).

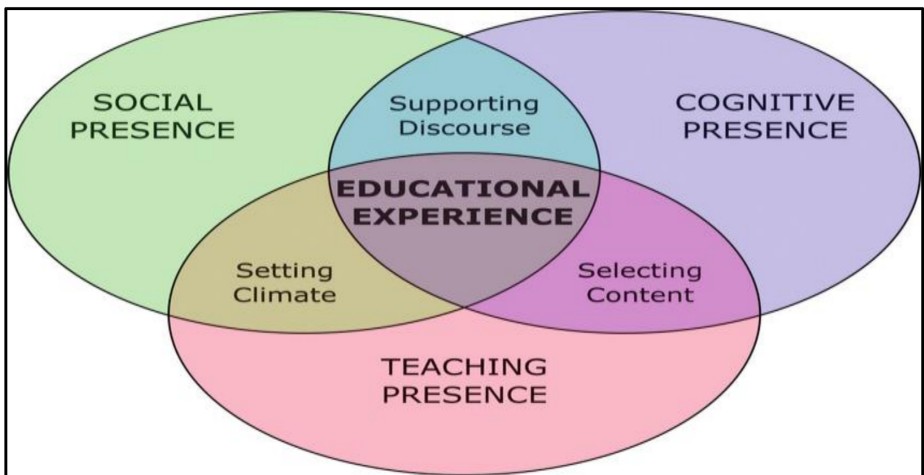

**Figure 1.** Community of Inquiry (CoI) adapted from Anderson, Rourke, Archer and Garrison [27].

This model, which illustrates the multi-faceted component of the teaching and learning process making up the educational experience, proposes that profound and significant learning occurs when three interdependent "presences" (cognitive, social, and teaching) interact [28].

The first element is the Cognitive Presence, which indicates that genuine learning can take place in a situation that underpins the advancement and development of basic reasoning abilities [25]. The Cognitive Presence is grounded in and characterised by the investigation of a specific substance; consequently, it works within the epistemological, social, and social articulation of the substance in a methodology that encourages the improvement of basic reasoning abilities [29,30]. The Cognitive Presence has been identified by other studies as Subjectification judgement [22] or Pragmatic/Personal knowledge-building [7]. These studies argue for importance of the Cognitive Presence in teaching and learning because it helps one to find and understand one's identity based on one's needs. Students learn by reflecting on their subjective experiences (subjectification) to produce actions (pragmatic). This suggests that students have more internal conscious intelligence to learn than external influence have on their intelligence. The cognitive presence should be part of any e-learning environment that helps students with visual impairment to find their identities, and learn from their strengths [23]. The cognitive presence positions individual needs and the situation/problem at the centre of e-learning. When students know and understand their identities, they will be able to draw from social and/or professional actions in addressing their needs. The external factors of e-learning consist of societal supporting discourse (social presence) and professional selecting content (teaching presence) needs, as the other two elements of presence [24,31].

The second element, the social presence, identifies with building a strong condition in which students experience an energetic level of well-being, being able to express their thoughts in a community-oriented setting. Such a setting would be capable of helping the students to acquire skills required to address societal needs. The social presence positions societal or community needs and local context at the centre of e-learning [20,32]. This suggests that Social Presence promotes collaboration or socialisation between students

and other stakeholders of e-learning [22,31]. Through the social presence, students can express contradictions, share perspectives, investigate contrasts, and acknowledge support and affirmation from friends and instructors. In other words, they are able to debate, in order to defend their positions within their local societies because they have acquired skills relevant to their societies. These skills have been developed through supportive discourse and the setting of a group climate in learning. Students with visual impairments use their collaborative skills to promote an e-learning environment sensitive to diversity, even if this means that they compromise international standards. As a result, an e-learning environment needs professional knowledge-building [7] or qualification judgement [22] in order to promote Teaching Presence.

The third and final element is the teaching presence that pays attention to the critical role of pedagogy, which deals with appropriate and tailored content for individual students within an environment that is conducive to learning. The teaching presence positions course content and professional needs at the centre of e-learning [33]. The main aim of the teaching presence is to follow professional, prescribed rules, producing a qualification in order to move to the next level of the activity [22]. The teaching presence focuses on setting the climate and selecting content to be prescribed for students and other stakeholders involved in e-learning. Although the teaching presence reflects a one-size-fit-all system of e-learning, it should always be supported by pragmatic e-learning that brings about student and academic identities [23,24].

The CoI, therefore, is deemed an appropriate model for promoting an inclusive e-learning environment that is conducive to facilitating educational experience for students with visual impairments. As such, this paper explores the use of CoI to facilitate the design of a holistic e-learning educational experience for students with visual impairments. The paper focuses on all elements that ensure effective interaction for successful teaching and learning, as well as student support, in which . . . the student becomes engaged by the content (whether people, text, images or other) and the technology disappears . . . [1].

The core of teaching and learning activities is the educational experience which, in this paper, relates to students with visual impairments. The importance of the educational experience is that it does not in any way negate or minimise the importance of the elements or their intersections. Rather, such teaching highlights the need for an all-encompassing experience for the student. For example, the teaching experience is about the teacher setting, the climate, and selecting content within a vacuum. However, all these elements work together with the other elements towards a holistic educational experience.

Anderson, Rourke, Archer, and Garrison [27] delineate three critical roles that a teacher performs in the process of creating an effective teaching presence. Firstly, teachers design and organise the learning experience, both before the establishment of the learning network, and amid its activity. According to Mpungose and Khoza [33]; Sokhulu [20], this role is called researcher reasoning, in which teachers research, thereby understanding their strengths to be incorporated in the design that drives e-learning. Secondly, instructing includes contriving and actualising exercises to energise conversation between and amongst students, between the educator and the students, and between individual students, gatherings of students, and substance assets [34]. According to Mpungose and Khoza [33]); Sokhulu [20], this role is named the socialisation or facilitation role. The socialisation role involves the identification of members of an e-learning environment, being able to group them, where necessary. Thirdly, the obligations of teaching go beyond directing the learning encounters when the teacher includes topic mastery through an assortment of types of direct guidance. The formation of an encouraging presence is not generally the sole undertaking of the formal instructor. In several specific situations, especially when teaching within senior spaces, an encouraging presence is assigned to or expected of students, who must contribute own abilities and information in creating the learning network or community.

## 3. Research Design and Methodology

This study used a pragmatic paradigm because it allows either qualitative or quantitative data or both to produce the actions. This paradigm is about observable teaching and learning actions based on individual needs [35]. This paradigm relates to the cognitive presence that combines social presence (qualitative issues) and teaching presence (quantitative issues) to produce teaching and learning actions [36,37]. The pragmatic paradigm was combined with critical discourse analysis (CDA) to analysis the findings of the four purposively and conveniently selected publications to be used as data for this study. In other words, this study has used the following four publications as data sources for this study, instead of using the views of participants/respondents.

The four selected studies used as data sources had rich information that addressed the following four research questions. The research questions support the use of CoI used this study to facilitate the design of a holistic e-learning experience for students with visual impairments:

(1) What are the learning needs of students with visual impairments? (2) How can students with visual impairments be taught in an ODeL environment? (3) How can the e-learning environment be designed to support students with visual impairments in times of pandemics which impose restrictions? (4) Why must students with visual impairments be supported and taught in particular ways?

Trustworthiness was addressed in terms of confirmability (neutrality—all the studies were conducted and published at UNISA and relevant to the purpose of the study); credibility (truth value—audit trail); dependability (consistency—direct quotations from the selected publications); transferability (applicability—by providing sufficient details of the relevant context). The limitation of the study was that it has to use the publications as the data sources. Academics were too busy to provide data because of COVID-19 and related national lockdown challenges. However, in the study, the authors of the publications were asked to read and confirm that the results were true reflections of what they had found in their studies. CDA is a process of engaging and critiquing text found in published sources. This study provided relevance in critiquing some four studies using CoI.

Guided analysis was used to analyse data which produced the findings presented based on themes. Guided analysis allowed CoI principles to become themes; while other themes were generated from the data of the four publications used as data sources.

## 4. Findings and Discussions

The findings are presented according to the themes. The study also substantiated the findings by means of discussions in order to re-contextualise them with relevant literature. This section of findings centres on these parts of presence, by defining and representing strategies to upgrade such presence. Recommendations are given in this section for successful inclusive instructor practice, particularly for the visually impaired students in an online-based learning setting.

### 4.1. Designing and Organising an Inclusive Online Learning Context (Teaching Presence)

The structure and development of inclusive course content, learning activities, and assessment systems establish the principal prospect for instructors to enhance the teaching presence. The instructor's role in shaping and preserving the course contents fluctuates from one working with materials and an instructional design made by others, to a "solitary officer" or educator who creates most of the content [5,6,26]. Irrespective of the instructor's formal role, online learning produces a prospect for adaptability and amendment of learning material that was not given by more established types of mediated educating and learning. The notable instructive and content assets of online teaching, with its ability to help a range of cooperation opportunities, allows the arrangement of content and a concomitant increment in independence and control) [20,28]. Instructors are never restricted to the development of "solid packages of learning" that cannot be effortlessly adjusted in light of students' needs [5,6]. In other words, the structure and association of activities in

the learning network can continue while the course is in progress. Such adaptability is not without expense, as customization of any item and content for the visually impaired students is more costly than large-scale institutionalised products. Consequently, e-learning teachers must make arrangements for the interspersal of learning tasks, or even material, to fulfil unique learning needs. As they turn out to be progressively educated members of formal training, these students are additionally requesting an increased contribution to the control of their learning [24,38]. In this adaptability and transaction of control, nevertheless, the need to invigorate, guide, and bolster learning for visually impaired students remains. These undertakings incorporate the plan of a progression of learning tasks that support autonomous investigation and community constructing, which profoundly investigate content knowledge, that offers various types of assessments and retort to normal and novel students' needs and goals [25].

The development of e-learning courses offers instructors opportunities to ingrain their own presence by setting up a customised and personalised meaning within the course content itself [23,33]. This presence is noted in enabling students to see individual energy and the instructor's enthusiasm for the subject. The main needs of students in this presence are positive attitudes of instructors towards students with visual impairments, understanding of disability policies and technologies like Braille [6]. When these needs are addressed, such will help students with visual impairments "to participate meaningfully in higher education and to achieve positive outcomes" [5] (p. 24). Oblinger [39] first expounded on a style of articulation, guided instructional communication, that presents content in a conversational rather than a scholastic style. This composition style encourages the student to distinguish a personalised interaction with the lecturer. Teaching methods, such as individual reflections, stories, and discourses of the educator's own battles and successes as he or she gained mastery of the subject content, might be uplifting, spurring many students on, while not being entirely inclusive. The design scope of the teaching presence includes the processes through which the instructor negotiates timelines for group activities and student project work. This is a critical coordinating and motivating purpose of formal online course design and development, and a primary means of setting and maintaining teaching presence. It is critical that course design adopt the universal design for learning (UDL) principles to enable all students to access e-learning platforms and content.

The developing and altering nature of content [26,39], as well as its presentation and other altered perspectives on content, are other basic actions related to teaching presence. Borrowing from teaching practices, the author envisions that the way in which instructive benchmarks for portraying, storing, and sequencing of instructive matter, and for formally exhibiting the way learning tasks are planned, will altogether change the structural role of numerous instructors. Roles will alter from material/content development to customization, application, and contextualization of learning successions, as described by Lehman and Conceicao [40]. Within such a large and diverse community, teaching presence incorporates the actions through which the instructor organises timetables for student project work and group activities. Teaching presence extends to basic organising and propelling the purpose of formal online course structure and development. Thus, an essential method for setting and maintaining teaching presence will include students of all backgrounds.

### 4.2. Online Application and the Mix of Models (Social Presence)

The cutting-edge online technologies underpin various media, every one of which can be consolidated into the plan of inclusive online learning courses. Such will foster the blend directly between prospects for the "synchronous and asynchronous interface and gathering and autonomous study activities remains a test, in any case" [41] (p. 19). There are two contending models of inclusive online learning, each with avid proponents and a developing assortment of research and hypothetical justifications for its viable application. The primary model, the community of learning model, utilises continuous synchronous

or nonconcurrent correspondence advancements towards virtual classrooms that are frequently displayed, both pedagogically and structurally, on the face-to-face classroom. This model has been developed from the phone-based audio (and later video and web) conferencing. Its advancement considers transmission of content to the student's office and home, bypassing costly remote forms of learning that were a component of the more seasoned virtual classroom models. The interplay of these two models has been observed as the contributing factors that supported UNISA students with visual impairments to participate meaningfully in their education [4–6,26]. Advocacy and Resource Centre for Students with Disabilities (ARCSWiD) was useful in driving online technologies for UNISA students with disabilities (SWD). This suggests the importance of educational development or technology Centres (ED/TCs) that need to be developed by educational institutions to support the unique educational needs of both staff and students [17]. The success of SWD depends on the usefulness of these centres that also promote useful conferencing for staff and students for the social presence [13,42,43].

Electronic computer-based conferencing frameworks consider nonconcurrent collaboration between and among lecturers and students. The synchronous virtual classroom is a comfortable instructive model with evident compatibility for educating and learning in face-to-face classrooms. It gives expanded access by spreading over the geographic scope; compels students and instructors to be available at a set single time. This issue is intensified when one class traverses multiple time zones. The unconventional adaptation of the virtual classroom transcends the global constraints yet can result in a deficiency of coordination. Students may then not feel "in a state of harmony" with the class [44] (p. 521). Structuring compulsion on inclusive online courses will progressively include reasonable determination of mixes of media and organizations. Such mixes will balance the differential limits of media to assist in creating social and cognitive presence with the variety of educational needs. The communications qualities required of specific material and content, and the cost, access, and "training prerequisites of the media will ultimately be a prerequisite" [45] (p. 248).

The other model of online learning includes self-directed learning (promoted by cognitive presence) in which students work independently and at their own pace through the course, under guidance. This model boosts adaptability; however, challenges the establishment instructor's ability to encourage social, or shared learning tasks. The autonomous/independent study model is quite often chosen in online learning models to permit uninterrupted enrolment or access to educational content. It is exceptionally onerous to advance communitarian learning or social exercises when students are studying altogether different s sections of the curriculum. The ongoing advancement of social software design [18,46,47] has roused some of us to start considering ways by which "unpaced" students can locate one another, take part in transient helpful activities, and generally create strong systems and study connections. Luckily, it is viable to consolidate synchronous, asynchronous, and independent study tasks in a single course. All these models have to be adapted to suit the needs of the students with visual impairments. As in the case of synchronous learning, normally the student would have to use voicenotes to respond to the chat session, or to ask someone else to read the comments so they can respond to the various threads. The presence of COVID-19 and the need for social distancing made it difficult for the students with visual impairments to receive support with synchronous learning from the Multipurpose Labs, or from their friends.

Throughout the years, in our own exchanges with online students, we have detected a profound gap between the individuals who desire the continuous correspondence and those who insist that they have online learning choices to maintain a strategic distance from the time imperatives forced upon them by synchronous or paced learning. Along these lines, numerous institutions are creating both paced and unpaced models, obliging students to select learning inclinations and requirements [33]. It is possible to offer discretionary synchronous exercises within a single class. Such approaches enable instructors to rapidly become more acquainted with the students from both an individual and expert perspective.

Instructors can then investigate their desires for the course, plot their own advantages in the subject, examine assessment exercises, and give the opportunity to all students to pose questions. Synchronous exercises are likewise helpful for visitor interviews for exceptional tasks, for example, discussions and introductions. Regardless of whether one's course structure or the accessible innovation blocks synchronous communication, there are still opportunities to infuse more than content-based lectures into the course. Inclusive online learning allows the instructor to work in video or sound introductions of themselves to improve their rapport with dispersed students, although this may not be entirely inclusive. In this way, the test for educators organising and spacing the online learning setting during COVID-19 and other pandemics, is to make a blend of learning exercises suitable to specific student needs, instructor abilities and style, learning goals of the programme of study, and institutional specialised limits. Accomplishing such within the ever-present budgetary limitations of formal instruction frameworks is a test that will coordinate Internet learning structure and usage for a long time to come.

### 4.3. Simplifying Inclusive Online Discourse (Cognitive Presence)

It is important to reflect that teacher presence has a fundamental duty to encourage discourse. In this paper, the authors utilise the term discourse as opposed to discussion. Discourse reflects the significance of identifying with the procedure or intensity of thinking [6,48], instead of the more social undertone of discussion. Again, discourse not only encourages the production of the community of inquiry, but it points to the methods by which students build their own perspectives. Students accomplish such by articulating their plans to other people through cognitive presence. Discourse, additionally, causes students to reveal misunderstandings in their own reasoning, or conflicts with the educator or other students. Such debates provide opportunities for the introduction to subjective dissonance which, from a "Piagetian" point of view, is essential to the intellectual development of most types of learners, apart from the deaf. In satisfying this part of cognitive presence, the instructor routinely peruses and reacts to students' commitments and concerns. Thus, the instructor always seeks approaches which help to understand the individual student, and the improvement of the learning community in general. Such can be achieved through reflecting on, in, and for their experiences [49,50].

The primary assignment for the e-learning instructor is to build a feeling of trust and well-being for the online community. Without this trust, students will feel awkward and compelled to post their cogitations and remarks. The author, for the most part, encourages this "trust arrangement" by having students post a progression of opening remarks about themselves. It is helpful to ask for explicit data, and to show a reply to the response request oneself. For instance, the online instructor may ask a student to explain the reason behind registering for the course, or their enthusiasm for the topic. This procedure has been used to effectively reach out towards the start of standard online synchronous sessions. Every student has been asked to react individually to a substance-related "question of the week" that sets the tone for the development of both social and cognitive presence. Various views afford thought-provoking development of social presence; for example, icebreakers [48]; various tasks obtained from adult learning and teaching tasks can be extremely successful in breaking down barriers to free and open discourse.

Currently, within many ODL institutions, several online courses depend widely on a model of discourse in which the instructor posts questions or aspects important to the readings or alternate types of content dissemination. The overreliance on this type of discourse becomes exhausting. Such discourse focuses largely on reacting to educator inquiry, as opposed to testing students, thereby defining their own questions and remarks about course content. The author has observed more noteworthy dimensions of participation, inspiration, and student fulfilment when discussions are driven by student moderators, as was observed by Rourke and Anderson [51]. Students cannot be expected to possess the essential aptitude to attempt an effective balance of class discussions; thus, role modelling by their instructor is characteristically useful. In an astute scrutiny of discourse, in non-

concurrent video- or computer-based conferencing, Rourke and Kanuka [52] take note of the deterrents felt by students in creating basic discourse. The researchers prescribe the requirement for "very much organized learning tasks with plainly characterized roles for instructors and students, and a technique for evaluating students' interest that mirrors the time and exertion required to take part in critical discourse" [52] (p. 105). Therefore, the proper management of online discourse may facilitate improved inclusive learning. Innovatively investing time and resources in an assortment of social programming instruments may be shown in both blended and online courses. Possibly the most used have been "blogs" or "webinars". While the degree to which these new instruments will hold favourable circumstances over more seasoned ones has not been clearly determined, they offer more benefit to people with visual impairments. There is little uncertainty that these new forms of discourses have created recharged enthusiasm for intelligent types of discourse to support inclusive online learning.

*4.4. Building a Teaching Presence to Support Inclusive Online Learning*

With reference to one specific model developed through the CoI principles, Salmon [53] (p. 11) suggests "a model for online moderators that differentiates the movement of assignments through which the online instructor travels during the time spent viably guiding an online course". The procedure starts by giving students both access and inspiration. At this stage, any specialised or social issues that repress investment are attended to, and students are urged to share information about themselves to make a virtual presence. In the second stage, Salmon recommends that online moderators develop web socialization by "building links between social and learning conditions" [53] (p. 26). In the third stage, the "information trade", Salmon recommends that the training assignment move to encourage learning undertakings, directing content-based exchanges, and uncovering students' confusions and false impressions. In the fourth stage, "learning development", students centre on achieving tasks that cooperatively and separately delineate their obvious comprehension, content, and methodologies. In the last "development" stage, students are ultimately in charge of their own learning and that of their group, by making final tasks, incrementally completing summative assignments, and exhibiting the accomplishment of learning results. These stages cannot be achieved when there are lockdown restrictions that do not allow students to have full access to assistive technologies and Internet connection for support.

Salmon's model offers a helpful guide and some planning apparatus for online learning instructors; in any case, it ought not to be viewed as rigid [9,17]. For instance, students might enter the online class having much specialised and social involvement with the Internet learning condition. In such cases, specialised and social issues may have been settled sometime prior. On the other hand, a heterogeneous group may have some exceptionally complex tech-savvy students and a few amateurs new to online-based learning conditions. Busy adult students might prefer to maintain a strategic distance from what they see as inefficient icebreakers related to Stages 1 and 2. Such students may wish to continue to the more substance-rich and possibly increasingly significant learning tasks related to later stages. In this manner, Salmon's model must be tailored to the unique needs of each online learning community. Inclusive teaching and learning may then become a reality after the needs of SWD have been identified.

## 5. Conclusions

This paper lays out the three noteworthy segments of teacher presence as one way of enhancing inclusive online learning. The study offers ideas and directions to strengthen the adequacy of the teaching function in online learning. The authors have not given a protracted rundown of "do's and don'ts" for inclusive online instructing. Rather, they have endeavoured to give a model concentrating on the three primary assignments of online instructors who reflect on, in and for their cognitive, social and teaching presence.

The setting of inclusive online learning is still evolving. The underlying innovations in technologies and the web itself are developing quickly into a second web, styled the

"semantic web" and a social web that is frequently called "Web 2.0" or "Fourth Industrial Revolution" (4IR) [54]. The 4IR implies the rapidly changing situations that disrupt the way we perform our actions or activities, and demand new, advanced technologies. The development of instructor and student specialists, the organising of content into learning objects, the social development and explanation of content by students, educators, and experts, and the formal articulation of learning associations are making a second-age web of 4IR technologies. These technologies offer both new capacities and difficulties to online instructors and students. So far, we are at the beginning period of the innovative and instructive advancement of web-based learning [55]. The central attributes of educating and adapting, in any case, and the three basic segments of showing presence—design and organization, facilitating discourse, and direct instruction—will remain critical parts of teaching effectiveness in both online learning and classroom guidance. At the centre of all these principles, instructor presence will be informed by students' needs. Student requirements will be generated through the self-reflections of both student and instructor, aimed at critiquing their actions to identify and understand their personal, social, and educational identities (Figure 2)

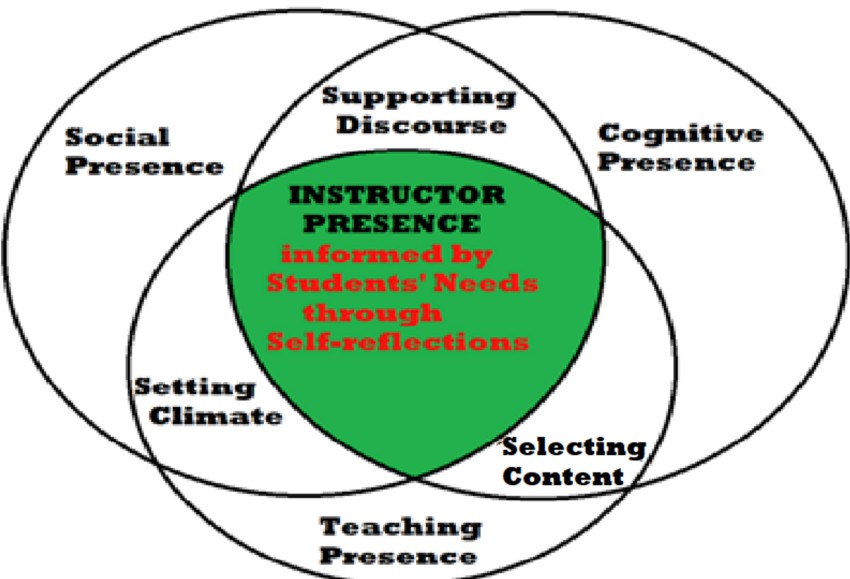

**Figure 2.** Instructor presence.

The community of inquiry provides information on the elements to be considered when integrating technology into teaching and learning, inclusion being at the centre of the learning-design activity. This means that those involved in the design process should have adequate knowledge and skills of how to design learning for greater accessibility. Such access to knowledge and skills could be acquired through gaining an understanding of a universal design for learning principles, derived from the community strata from which all students originate. Special attention should be paid to ensuring that learning design also becomes accessible during pandemics. Perhaps there should be a follow-up study on how students with impairments experience the Advocacy and Resource Centre for Students with Disabilities (ARCSWiD) or ED/TCs used by their institutions to support their education.

**Author Contributions:** Conceptualisation, S.A.N.-M. and S.B.K.; methodology, S.B.K.; software, S.A.N.-M. and S.B.K.; validation, S.A.N.-M. and S.B.K.; formal analysis, S.A.N.-M. and S.B.K.; investigation, S.A.N.-M.; resources, S.A.N.-M. and S.B.K.; data curation, S.B.K.; writing—original draft preparation, S.A.N.-M.; writing—review and editing, S.A.N.-M. and S.B.K.; visualisation, S.B.K.; supervision, S.B.K.; project administration, S.A.N.-M.; funding acquisition, S.A.N.-M. All authors have read and agreed to the published version of the manuscript.

**Funding:** This research received no external funding.

**Institutional Review Board Statement:** Not applicable.

**Informed Consent Statement:** Not applicable.

**Data Availability Statement:** Published articles used in this study are included in the list of references.

**Conflicts of Interest:** The authors declare no conflict of interest.

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
