# Peer review of "Using Community of Inquiry (CoI) to Facilitate the Design of a Holistic E-Learning Experience for Students with Visual Impairments"

_education, doi:10.3390/educsci11040152_

Round 1
Reviewer 1 Report
After analyzing the manuscript "Using Community of Inquiry (CoI) to Facilitate the Design of a Holistic E-learning Experience for Students with Visual Impairments" there are several aspects I should mention. Firstly, the manuscript does not follow a coherent structure or a "standard" structure for research articles. Nowhere is the research method developed specified (which is necessary if the study is to be replicated). Furthermore, the theoretical framework mostly refers to authors prior to 2015. This aspect is not understandable. Given that the subject matter presented in the article has many manuscripts from recent years in WoS. Furthermore, the references do not follow the recommended structure for MDPI. In short, it is a manuscript that shows educational practices, but at no point does it analyze, in an appropriate manner, whether these practices are adequate or not. It does not have a structure to be considered a research article.
Author Response
We have revised the article to follow a standard structure for research articles (Introduction, Literature review, Research Design with Methodology, Findings with Discussions, Conclusion, and References).
Research Design with Methodology is included (pp. 4-5)
Studies that are younger than 5 years old were included in the revised article.
References follow the editorial recommended format.
Educational practices are included under the Findings and Discussions (from p. 5…).
Reviewer 2 Report
It was a nice to read this paper as this is a very important topic, nonetheless in order to make the paper publishable I would ask you to include the following changes:
- Please separate “Introduction” (very first lines) from “Literature review( from 1.1.).
- The submission lacks research questions that are essential in submissions, based on their qualitative or quantitative approach. In the Introduction section please introduce your research questions (as this is a qualitative study).
- In the Introduction section please put more emphasis on the novelty of this paper, hat new is brought by this paper to the literature?
- I would start writing the introduction from the describing the university, that I would move to more empirical part, I would definitively start the introduction from the problem identification. Please re-shape the introduction in order to show the research value of the paper for readers in your introduction. Please see a good example in ‘Educational Sciences’ in the following article at https://doi.org/10.3390/educsci10030062
- In the Literature Review section try to make a short summary what are the current research (previous studies) in this field and what is your contribution, how you will continue the extant literature.
- In my opinion the submission does not present required contribution. Please show what is your contribution?
- The method used to achieve results is far too general, lacking the necessary scientific rigor. Please try to improve it, at least in the description.
- In the Conclusions please remember that we need to have four compulsory elements: (i) Research Limitations and (ii) general summary of empirical results, but also – what I miss – (iii) implications for practice and (iv) suggestions for further research in this topic. I really miss some of them, e.g. limitations.
Minor comments:
- All abbreviations should be explained, even if obvious, e.g. ODL in line 7 (you explained ODeL line 27 or ICT line 6).
I hope these comments will be useful for you. Good luck with revising your paper.
Author Response
The introduction has been separated from the literature using relevant headings.
Research questions were included (what are the learning needs of students with visual impairments? How can students with visual impairments be taught in an ODeL environment? How can the e-learning environment be designed to support students with visual impairments in times of pandemics which impose restrictions? Why do students with visual impairments supported and taught in particular ways?) (p. 2 and p. 5).
The novelty of the paper has been added as recommended both in the introduction and literature review sections.
The introduction has been revised according to the recommendations and the example given.
The literature review has been revised as recommended to be an independent section separated from the introduction.
The recommended sections (research limitations – under methodology, findings, practices, recommended future research – under conclusion…) were added in the revised article.
All abbreviations were explained.
Reviewer 3 Report
I consider that the article is correctly structured, has an adequate theoretical foundation and represents progress.
Author Response
Noted with thanks.
Reviewer 4 Report
Overall I found this paper interesting but I felt that it added little to the existing literature. A large proportion of the paper is dedicated to providing summaries of existing models (COI, SDL, Salomon, and others).
While these models are clearly important and pertinent to the educational needs of students with visual impairments, this paper does not add to the literature in terms of making novel connections.
Pityana ODeL model played a prominent role in the introduction but I could not locate the cited paper/speech (2009). ODL and ODeL seem to be used interchangeably. Given the prominence of the acronyms, it might be worth delineating and choosing one for the paper. Unless you are trying to make distinctions w/ the different sections. I suspect the Pityana paper would clarify that for the reader.
The organization of the paper needs improvement. A clear statement of purpose and a logical progression of the topics would benefit the reader.
Author Response
The article has used figure 2 to show the importance of Instructor Presence as the contribution towards the existing literature and show some gaps in these models.
The models were critiqued to produce novel connection in Figure 2.
ODeL is the extension of ODL and we have revised the section on this in order to clarify it.
The article has been revised and improved according to the recommendations of the reviewers.
Reviewer 5 Report
The manuscript presents a promising theoretical approach to the use of CoI for students with visual impairments. The aim of the study is clearly stated by the author and focuses on an educational problem that is being studied internationally (need for more inclusive e-learning environments). However, The literature review is weak, it does not take into account the latest studies in the field of digital teaching competence or articles published. Most of the references cited are from 1990 to 2003, and I miss more practical examples where IoT are used specifically for helping students with visual impairments.
There are also some typographic errors (see "social, and social" in line 96 for example).
Finally, in the conclusion section, author states that this study has focused on giving an expansive model that could be useful, but without explaining the particular aspects or parts of this model (maybe the use of a figure could help). It is also necessary to clearly state for whom the investigation is important, which educational levels and type of students and teachers could be interested by this model. Finally, which are the practical applications and implications of the paper?
Author Response
The literature review has been revised as recommended to include current studies that are younger than 5 years old.
The practical examples were included in the Findings and Discussion section.
The article has been edited by professional editor to correct technical and typographic errors.
Figure 2 was included to explain the model as recommended.
The article is based on UNISA students with impairments issues. Therefore, it may be useful to university students. The findings were from UNISA published studies on these students.
The article may be useful to academics to understand the Instructor Presence as a drive for education.
Reviewer 6 Report
Thank you for presenting key frameworks useful to guiding online learning. The discussion helps to shift the emphasis away from the technologies and content to the process of learning. To strengthen the paper, please consider the following:
1) A greater emphasis on the needs of students with visual impairments. Integrate throughout the paper links back to what is the core emphasis in the paper i.e. online learning for those with visual impairments. While I can see the relevance of what is being said to students with visual impairments, the reader is left somewhat to themselves to make these connections.
2) Particularly in the introduction line 138-139 it says "The Col is therefore..." Consider making a clearer argument or more explicit argument as to why Col is an appropriate model. Perhaps some literature around the needs of visually impaired learners may be useful here.
3) UDL is mentioned, but given the emphasis in on students with a visual impairment, elaborate more deeply on the principles of UDL; consider showing how these principles link to the other frameworks such as COL and discourse. One approach may be to introduce UDL first and proceed by weaving the principles of UDL together with the other valuable ideas you bring to the fore.
4) While it is clear that the paper is not intended to present do's and don't, seek to bring the many valuable ideas presented throughout the paper together at the conclusion and make an explicit link back to the needs of visually impaired students as well as others.
As it stands the paper is a nice overview of some very important concepts in the design of online learning, but consider bringing more focus to the apparent core of the paper around visually impaired students.
Best wishes.
Author Response
Students’ needs have been emphasised in difference sections especially on page 6. We also integrated throughout the article the links to the purpose of the study.
We have added relevant literature that clarity the needs of visually impaired students.
We have clarified the issue of ODL as extended to ODeL and how UNISA as UDL or ODL institution uses the principles of CoI.
We have revised the article to focus on the core of the article (students with visual impairments) as recommended.
Round 2
Reviewer 1 Report
Dear authors,
I note major changes in your manuscript. I only ask you to organize section 3. You should not put the references as they are in that section, only the citations.
Good work on the proposed changes.
Author Response
Reviewer comment: I note major changes in your manuscript. I only ask you to organize section 3. You should not put the references as they are in that section, only the citations.
Author comment: The wrongly placed references have been removed.
Reviewer comment: Good work on the proposed changes.
Author comments: Thank you for your positive feedback.
Reviewer 2 Report
Dear Author,
Thank you for seding a revised version of your article. I would strongly to include two minor revisions:
1) Please make your introduction longer: a) why this topic is worth studying, what is a novelty/oroginality if this article, b) what is the objective of the article and the research questions (RQ1, RQ2 ..... etc), c) what methods were used, d) what is the expected contribution of this article
2) try to use more scientific references in your article (3-5 more sources), e.g.
Tarabasz, A., Selaković, M., & Abraham, C. (2018). The Classroom of the Future: Disrupting the Concept of Contemporary Business Education. Entrepreneurial Business and Economics Review, 6(4), 231-245. https://doi.org/10.15678/EBER.2018.060413
Good luck with the final decision, as now I see the need for only minor changes.
Author Response
Reviewer comment:
1. Please make your introduction longer: a) why this topic is worth studying, what is a novelty/oroginality if this article, b) what is the objective of the article and the research questions (RQ1, RQ2 ..... etc), c) what methods were used, d) what is the expected contribution of this article
Author correction: UNISA, thus, has introduced innovative models of teaching and learning as part of its student-support initiatives, which entail the use of online learning platforms. However, the developed models lack the identification of those whose needs are addressed by these models (Maboe, Eloff, & Schoeman, 2018). As a result, the needs of students with disabilities are not considered in most of the ODeL models, because they are unknown (Ntombela, 2020; Satar, 2019). Therefore, this study explores the use of the community of inquiry (CoI) to facilitate the design of a holistic e-learning experience for students with visual impairments. It was guided by the following research questions: (1) What are the learning needs of students with visual impairments? (2) How can students with visual impairments be taught in an ODeL environment? (3) How can the e-learning environment be designed to support students with visual impairments in times of pandemics which impose restrictions? (4) Why must students with visual impairments be supported and taught in particular ways? It is hoped that this paper will contribute towards better use of CoI to mediate positive learning experiences for students with disabilities.
2) try to use more scientific references in your article (3-5 more sources), e.g.
Tarabasz, A., Selaković, M., & Abraham, C. (2018). The Classroom of the Future: Disrupting the Concept of Contemporary Business Education. Entrepreneurial Business and Economics Review, 6(4), 231-245. https://doi.org/10.15678/EBER.2018.060413
Author comments: the authors have used scientific references that are relevant to the topic at hand. We humbly request to maintain the references we have employed thus far.
Reviewer 4 Report
The authors have responded thoughtfully to reviewer suggestions. The purpose and methods are now more clear.
Author Response
Reviewer response: The authors have responded thoughtfully to reviewer suggestions. The purpose and methods are now more clear.
Author response: Thank you for your kind comment. All comments have been addressed for this review.
Reviewer 5 Report
The changes in the manuscript have given this study a most solid background and theoretical foundation, and the methodology and conclusions are clearly stated. Lines 197 to 213 in page 5 should be in the reference section.
Author Response
Reviewer comment: The changes in the manuscript have given this study a most solid background and theoretical foundation, and the methodology and conclusions are clearly stated.
Author comment: Thank you for your positive feedback
Reviewer comment: Lines 197 to 213 in page 5 should be in the reference section.
Author comment: These references have been removed and placed in the reference section.